# Wasserstein Training of Restricted Boltzmann Machines

**Grégoire Montavon**
Technische Universität Berlin
gregoire.montavon@tu-berlin.de

**Klaus-Robert Müller**[*]
Technische Universität Berlin
klaus-robert.mueller@tu-berlin.de

**Marco Cuturi**
CREST, ENSAE, Université Paris-Saclay
marco.cuturi@ensae.fr

## Abstract

Boltzmann machines are able to learn highly complex, multimodal, structured and multiscale real-world data distributions. Parameters of the model are usually learned by minimizing the Kullback-Leibler (KL) divergence from training samples to the learned model. We propose in this work a novel approach for Boltzmann machine training which assumes that a meaningful metric between observations is known. This metric between observations can then be used to define the Wasserstein distance between the distribution induced by the Boltzmann machine on the one hand, and that given by the training sample on the other hand. We derive a gradient of that distance with respect to the model parameters. Minimization of this new objective leads to generative models with different statistical properties. We demonstrate their practical potential on data completion and denoising, for which the metric between observations plays a crucial role.

## 1 Introduction

Boltzmann machines [1] are powerful generative models that can be used to approximate a large class of real-world data distributions, such as handwritten characters [9], speech segments [7], or multimodal data [16]. Boltzmann machines share similarities with neural networks in their capability to extract features at multiple scales, and to build well-generalizing hierarchical data representations [15, 13]. The restricted Boltzmann machine (RBM) is a special type of Boltzmann machine composed of one layer of latent variables, and defining a probability distribution $p_\theta(\boldsymbol{x})$ over a set of $d$ binary observed variables whose state is represented by the binary vector $\boldsymbol{x} \in \{0,1\}^d$, and with a parameter vector $\theta$ to be learned.

Given an empirical probability distribution $\hat{p}(\boldsymbol{x}) = \frac{1}{N} \sum_{n=1}^{N} \delta_{\boldsymbol{x}_n}$ where $(\boldsymbol{x}_n)_n$ is a list of $N$ observations in $\{0,1\}^d$, an RBM can be trained using information-theoretic divergences (see for example [12]) by minimizing with respect to $\theta$ a divergence $\Delta(\hat{p}, p_\theta)$ between the sample empirical measure $\hat{p}$ and the modeled distribution $p_\theta$. When $\Delta$ is for instance the KL divergence, this approach results in the well-known Maximum Likelihood Estimator (MLE), which yields gradients for the $\theta$ of the form

$$\nabla_\theta \mathrm{KL}(\hat{p} \,\|\, p_\theta) = -\frac{1}{N} \sum_{n=1}^{N} \nabla_\theta \log p_\theta(\boldsymbol{x}_n) = -\big\langle \nabla_\theta \log p_\theta(\boldsymbol{x}) \big\rangle_{\hat{p}}, \qquad (1)$$

where the bracket notation $\langle \cdot \rangle_p$ indicates an expectation with respect to $p$. Alternative choices for $\Delta$ are the Bhattacharrya/Hellinger and Euclidean distances between distributions, or more generally

---

[*]Also with the Department of Brain and Cognitive Engineering, Korea University.

$F$-divergences or $M$-estimators [10]. They all result in comparable gradient terms, that try to adjust $\theta$ so that the fitting terms $p_\theta(\boldsymbol{x}_n)$ grow as large as possible.

We explore in this work a different scenario: what if $\theta$ is chosen so that $p_\theta(\boldsymbol{x})$ is large, on average, when $\boldsymbol{x}$ is *close* to a data point $\boldsymbol{x}_n$ in some sense, but not necessarily when $\boldsymbol{x}$ coincides *exactly* with $\boldsymbol{x}_n$? To adopt such a geometric criterion, we must first define what closeness between observations means. In almost all applications of Boltzmann machines, such a metric between observations is readily available: One can for example consider the Hamming distance between binary vectors, or any other metric motivated by practical considerations[2]. This being done, the geometric criterion we have drawn can be materialized by considering for $\Delta$ the Wasserstein distance [20] (a.k.a. the Kantorovich or the earth mover's distance [14]) between measures. This choice was considered in theory by [2], who proved its statistical consistency, but was never considered practically to the best of our knowledge. This paper describes a practical derivation for a *minimum Kantorovich distance estimator* [2] for Boltzmann machines, which can scale up to tens of thousands of observations. As will be described in this paper, recent advances in the fast approximation of Wasserstein distances [5] and their derivatives [6] play an important role in the practical implementation of these computations. Before describing this approach in detail, we would like to insist that measuring goodness-of-fit with the Wasserstein distance results in a considerably different perspective than that provided by a Kullback-Leibler/MLE approach. This difference is illustrated in Figure 1, where a probability $p_\theta$ can be close from a KL perspective to a given empirical measure $\hat{p}$, but far from the same measure $p$ in the Wasserstein sense. Conversely, a different probability $p_{\theta'}$ can miss the mark from a KL viewpoint but achieve a low Wasserstein distance to $\hat{p}$. Before proceeding to the rest of this paper, let us mention that Wasserstein distances have a broad appeal for machine learning. That distance was for instance introduced in the context of supervised inference by [8], who used it to compute a predictive *loss* between the output of a multilabel classifier against its ground truth, or for domain adaptation, by [4].

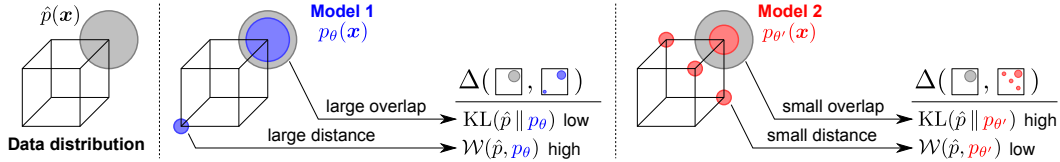

Figure 1: Empirical distribution $\hat{p}(\boldsymbol{x})$ (gray) defined on the set of states $\{0,1\}^d$ with $d=3$ superposed to two possible models of it defined on the same set of states. The size of the circles indicates the probability mass for each state. For each model, we show its KL and Wasserstein divergences from $\hat{p}(\boldsymbol{x})$, and an explanation of why the divergences are low or high: a large/small overlap with $\hat{p}(\boldsymbol{x})$, or a large/small distance from $\hat{p}(\boldsymbol{x})$.

## 2 Minimum Wasserstein Distance Estimation

Consider two probabilities $p, q$ in $\mathcal{P}(\mathcal{X})$, the set of probabilities on $\mathcal{X} = \{0,1\}^d$. Namely, two maps $p, q$ in $\mathbb{R}_+^{\mathcal{X}}$ such that $\sum_{\boldsymbol{x}} p(\boldsymbol{x}) = \sum_{\boldsymbol{x}} q(\boldsymbol{x}) = 1$, where we omit $\boldsymbol{x} \in \mathcal{X}$ under the summation sign. Consider a cost function defined on $\mathcal{X} \times \mathcal{X}$, typically a distance $D : \mathcal{X} \times \mathcal{X} \to \mathbb{R}_+$. Given a constant $\gamma \geq 0$, the $\gamma$-smoothed Wasserstein distance [5] is equal to

$$\mathcal{W}_\gamma(p,q) = \min_{\pi \in \Pi(p,q)} \langle D(\boldsymbol{x},\boldsymbol{x}')\rangle_\pi - \gamma H(\pi), \tag{2}$$

where $\Pi(p,q)$ is the set of joint probabilities $\pi$ on $\mathcal{X} \times \mathcal{X}$ such that $\sum_{\boldsymbol{x}'} \pi(\boldsymbol{x},\boldsymbol{x}') = p(\boldsymbol{x})$, $\sum_{\boldsymbol{x}} \pi(\boldsymbol{x},\boldsymbol{x}') = q(\boldsymbol{x}')$ and $H(\pi) = -\sum_{\boldsymbol{x}\boldsymbol{x}'} \pi(\boldsymbol{x},\boldsymbol{x}') \log \pi(\boldsymbol{x},\boldsymbol{x}')$ is the Shannon entropy of $\pi$. This optimization problem, a strictly convex program, has an equivalent dual formulation [6] which involves instead two real-valued functions $\alpha, \beta$ in $\mathbb{R}^{\mathcal{X}}$ and which plays an important role in this paper:

$$\mathcal{W}_\gamma(p,q) = \max_{\alpha,\beta \in \mathbb{R}^{\mathcal{X}}} \langle \alpha(\boldsymbol{x})\rangle_p + \langle \beta(\boldsymbol{x}')\rangle_q - \gamma \sum_{\boldsymbol{x}\boldsymbol{x}'} e^{\frac{1}{\gamma}(\alpha(\boldsymbol{x})+\beta(\boldsymbol{x}')-D(\boldsymbol{x},\boldsymbol{x}'))-1}. \tag{3}$$

**Smooth Wasserstein Distances**  The "true" Wasserstein distance corresponds to the case where $\gamma = 0$, that is when Equation (2) is stripped of the entropic term. One can easily verify that that definition matches the usual linear program used to describe Wasserstein/EMD distances [14]. When $\gamma \to 0$ in Equation (3), one also recovers the Kantorovich dual formulation, because the rightmost regularizer converges to the indicator function of the feasible set of the dual optimal transport problem, $\alpha(\boldsymbol{x}) + \beta(\boldsymbol{x}') \leq D(\boldsymbol{x}, \boldsymbol{x}')$. We consider in this paper the case $\gamma > 0$ because it was shown in [5] to considerably facilitate computations, and in [6] to result in a divergence $\mathcal{W}_\gamma(p, q)$ which, unlike the case $\gamma = 0$, is *differentiable* w.r.t to the first variable. Looking at the dual formulation in Equation (3), one can see that this gradient is equal to $\alpha^\star$, the centered optimal dual variable (the centering step for $\alpha^\star$ ensures the orthogonality with respect to the simplex constraint).

Sensitivity analysis gives a clear interpretation to the quantity $\alpha^\star(\boldsymbol{x})$: It measures the *cost* for each unit of mass placed by $p$ at $\boldsymbol{x}$ when computing the Wasserstein distance $\mathcal{W}_\gamma(p, q)$. To decrease $\mathcal{W}_\gamma(p, q)$, it might thus be favorable to transfer mass in $p$ from points where $\alpha(\boldsymbol{x})$ is high to place it on points where $\alpha(\boldsymbol{x})$ is low. This idea can be used, by a simple application of the chain rule, to minimize, given a fixed target probability $p$, the quantity $\mathcal{W}_\gamma(p_\theta, p)$ with respect to $\theta$.

**Proposition 1.** *Let $p_\theta(\boldsymbol{x}) = \frac{1}{Z} e^{-F_\theta(\boldsymbol{x})}$ be a parameterized family of probability distributions where $F_\theta(\boldsymbol{x})$ is a differentiable function of $\theta \in \Theta$ and we write $G_\theta = \langle \nabla_\theta F_\theta(\boldsymbol{x}) \rangle_{p_\theta}$. Let $\alpha^\star$ be the centered optimal dual solution of $\mathcal{W}_\gamma(p_\theta, p)$ as described in Equation (3). The gradient of the smoothed Wasserstein distance with respect to $\theta$ is given by*

$$\nabla_\theta \mathcal{W}_\gamma(p_\theta, p) = \langle \alpha^\star(\boldsymbol{x}) \rangle_{p_\theta} G_\theta - \langle \alpha^\star(\boldsymbol{x}) \nabla_\theta F_\theta(\boldsymbol{x}) \rangle_{p_\theta}. \tag{4}$$

*Proof.*  This result is a direct application of the chain rule: We have

$$\nabla_\theta \mathcal{W}_\gamma(p_\theta, p) = \left( \frac{\partial p_\theta}{\partial \theta} \right)^T \frac{\partial \mathcal{W}_\gamma(p_\theta, q)}{\partial p_\theta}.$$

As mentioned in [6], the rightmost term is the optimal dual variable (the Kantorovich potential) $\partial \mathcal{W}_\gamma(p_\theta, q)/\partial p_\theta = \alpha^\star$. The Jacobian $(\partial p_\theta / \partial \theta)$ is a linear map $\Theta \to \mathcal{X}$. For a given $\boldsymbol{x}'$,

$$\partial p_\theta(\boldsymbol{x}')/\partial \theta = p_\theta(\boldsymbol{x}') G_\theta - \nabla F_\theta(\boldsymbol{x}') p_\theta(\boldsymbol{x}').$$

As a consequence, $\left( \frac{\partial p_\theta}{\partial \theta} \right)^T \alpha^\star$ is the integral w.r.t. $\boldsymbol{x}'$ of the term above multiplied by $\alpha^\star(\boldsymbol{x}')$, which results in Equation (4). □

**Comparison with the KL Fitting Error**  The target distribution $p$ plays a direct role in the formation of the gradient of $\mathrm{KL}(\hat{p} \| p_\theta)$ w.r.t. $\theta$ through the term $\langle \nabla_\theta F_\theta(\boldsymbol{x}) \rangle_p$ in Equation (1). The Wasserstein gradient incorporates the knowledge of $p$ in a different way, by considering, on the support of $p_\theta$ only, points $\boldsymbol{x}$ that correspond to high potentials (costs) $\alpha(\boldsymbol{x})$ when computing the distance of $p_\theta$ to $p$. A high potential at $\boldsymbol{x}$ means that the probability $p_\theta(\boldsymbol{x})$ should be lowered if one were to decrease $\mathcal{W}_\gamma(p_\theta, p)$, by varying $\theta$ accordingly.

**Sampling Approximation**  The gradient in Equation (4) is intractable, since it involves solving an optimal (smoothed) transport problem over probabilities defined on $2^d$ states. In practice, we replace expectations w.r.t $p_\theta$ by an empirical distribution formed by sampling from the model $p_\theta$ (e.g. the PCD sample [18]). Given a sample $(\widetilde{\boldsymbol{x}}_n)_n$ of size $\widetilde{N}$ generated by the model, we define $\hat{p}_\theta = \sum_{n=1}^{\widetilde{N}} \delta_{\widetilde{\boldsymbol{x}}_n} / \widetilde{N}$. The tilde is used to differentiate the sample generated by the model from the empirical observations. Because the dual potential $\alpha^\star$ is centered and $\hat{p}_\theta$ is a measure with uniform weights, $\langle \alpha^\star(\boldsymbol{x}) \rangle_{\hat{p}_\theta} = 0$ which simplifies the approximation of the gradient to

$$\widehat{\nabla}_\theta \mathcal{W}_\gamma(p_\theta, \hat{p}) = -\frac{1}{\widetilde{N}} \sum_{n=1}^{\widetilde{N}} \hat{\alpha}^\star(\widetilde{\boldsymbol{x}}_n) \nabla_\theta F_\theta(\widetilde{\boldsymbol{x}}_n) \tag{5}$$

where $\hat{\alpha}^\star$ is the solution of the discrete smooth Wasserstein dual between the two empirical distributions $\hat{p}$ and $\hat{p}_\theta$, which have respectively supports of size $N$ and $\widetilde{N}$. In practical terms, $\hat{\alpha}^\star$ is a vector of size $\widetilde{N}$, one coefficient for each PCD sample, which can be computed by following the algorithm below [6]. To keep notations simple, we describe it in terms of generic probabilities $p$ and $q$, having in mind these are in practice the training and simulated empirical measures $\hat{p}$ and $\hat{p}_\theta$.

**Computing $\alpha^\star$** When $\gamma > 0$, the optimal variable $\alpha^\star$ corresponding to $\mathcal{W}_\gamma(p,q)$ can be recovered through the Sinkhorn algorithm with a cost which grows as the product $|p||q|$ of the sizes of the support of $p$ and $q$, where $|p| = \sum_{\boldsymbol{x}} 1_{p(\boldsymbol{x})>0}$. The algorithm is well known but we adapt it here to our setting, see [6, Alg.3] for a more precise description. To ease notations, we consider an arbitrary ordering of $\mathcal{X}$, a set of cardinal $2^d$, and identify its elements with indices $1 \le i \le 2^d$. Let $I = (i_1, \cdots, i_{|p|})$ be the ordered family of indices in the set $\{i \,|\, p(i) > 0\}$ and define $J$ accordingly for $q$. $I$ and $J$ have respective lengths $|p|$ and $|q|$. Form the matrix $K = [e^{-D(i,j)/\gamma}]_{i \in I, j \in J}$ of size $|p|$ and $|q|$. Choose now two positive vectors $u \in \mathbb{R}^{|p|}_{++}$ and $v \in \mathbb{R}^{|q|}_{++}$ at random, and repeat until $u, v$ converge in some metric the operations $u \leftarrow p/(Kv), v \leftarrow q/(K^T u)$. Upon convergence, the optimal variable $\alpha^\star$ is zero everywhere except for $\alpha^\star(i_a) = \log(u_a/\tilde{u})/\gamma$ where $1 \le a \le |p|$ and $\tilde{u}$ is the geometric mean of vector $u$ (which ensures that $\alpha^\star$ is centered).

## 3    Application to Restricted Boltzmann Machines

The restricted Boltzmann machine (RBM) is a generative model of binary data that is composed of $d$ binary observed variables and $h$ binary explanatory variables. The vector $\boldsymbol{x} \in \{0,1\}^d$ represents the state of observed variables, and the vector $\boldsymbol{y} \in \{0,1\}^h$ represents the state of explanatory variables. The RBM associates to each configuration $\boldsymbol{x}$ of observed variables a probability $p_\theta(\boldsymbol{x})$ defined as $p_\theta(\boldsymbol{x}) = \sum_{\boldsymbol{y}} e^{-E_\theta(\boldsymbol{x},\boldsymbol{y})}/Z_\theta$, where $E_\theta(\boldsymbol{x},\boldsymbol{y}) = -\boldsymbol{a}^T \boldsymbol{x} - \sum_{j=1}^h y_j(\boldsymbol{w}_j^T \boldsymbol{x} + b_j)$ is called the energy and $Z_\theta = \sum_{\boldsymbol{x},\boldsymbol{y}} e^{-E_\theta(\boldsymbol{x},\boldsymbol{y})}$ is the partition function that normalizes the probability distribution to one. The parameters $\theta = (\boldsymbol{a}, \{\boldsymbol{w}_j, b_j\}_{j=1}^h)$ of the RBM are learned from the data. Knowing the state $\boldsymbol{x}$ of the observed variables, the explanatory variables are independent Bernoulli-distributed with $\Pr(y_j = 1|\boldsymbol{x}) = \sigma(\boldsymbol{w}_j^T \boldsymbol{x} + b_j)$, where $\sigma$ is the logistic map $z \mapsto (1 + e^{-z})^{-1}$. Conversely, knowing the state $\boldsymbol{y}$ of the explanatory variables, the observed variables on which the probability distribution is defined can also be sampled independently, leading to an efficient alternate Gibbs sampling procedure for $p_\theta$. In this RBM model, explanatory variables can be analytically marginalized, which yields the following probability model:
$$p_\theta(\boldsymbol{x}) = e^{-F_\theta(\boldsymbol{x})}/Z'_\theta,$$
where $F_\theta(\boldsymbol{x}) = -\boldsymbol{a}^T \boldsymbol{x} - \sum_{j=1}^h \log(1 + \exp(\boldsymbol{w}_j^T \boldsymbol{x} + b_j))$ is the free energy associated to this model and $Z'_\theta = \sum_{\boldsymbol{x}} e^{-F_\theta(\boldsymbol{x})}$ is the partition function.

**Wasserstein Gradient of the RBM** Having written the RBM in its free energy form, the Wasserstein gradient can be obtained by computing the gradient of $F_\theta(\boldsymbol{x})$ and injecting it in Equation (5):
$$\widehat{\nabla}_{\boldsymbol{w}_j} \mathcal{W}_\gamma(\hat{p}, p_\theta) = \langle \alpha^\star(\boldsymbol{x})\, \sigma(z_j)\, \boldsymbol{x} \rangle_{\hat{p}_\theta},$$
where $z_j = \boldsymbol{w}_j^T \boldsymbol{x} + b_j$. Gradients with respect to parameters $\boldsymbol{a}$ and $\{b_j\}_j$ can also be obtained by the same means. In comparison, the gradient of the KL divergence is given by
$$\widehat{\nabla}_{\boldsymbol{w}_j} \text{KL}(\hat{p} \,\|\, p_\theta) = \langle \sigma(z_j)\, \boldsymbol{x} \rangle_{\hat{p}_\theta} - \langle \sigma(z_j)\, \boldsymbol{x} \rangle_{\hat{p}}.$$

While the Wasserstein gradient can in the same way as the KL gradient be expressed in a very simple form, the first one is not sum-decomposable. A simple manifestation of the non-decomposability occurs for $\widetilde{N} = 1$ (smallest possible sample size): In that case, $\alpha(\widetilde{\boldsymbol{x}}_n) = 0$ due to the centering constraint (see Section 2), thus making the gradient zero.

**Stability and KL Regularization** Unlike the KL gradient, the Wasserstein gradient depends only indirectly on the data distribution $\hat{p}$. This is a problem when the sample $\hat{p}_\theta$ generated by the model strongly differs from the examples coming from $\hat{p}$, because there is no weighting $(\alpha(\widetilde{\boldsymbol{x}}_n))_n$ of the generated sample that can represent the desired direction in the parameter space $\Theta$. In that case, the Wasserstein gradient will point to a bad local minimum. Closeness between the two empirical samples from this optimization perspective can be ensured by adding a *regularization* term to the objective that incorporates both the usual quadratic containment term, but also the KL term, that forces proximity to $\hat{p}$ due to the direct dependence of its gradient on it. The optimization problem becomes:
$$\min_{\theta \in \Theta} \mathcal{W}_\gamma(\hat{p}, p_\theta) + \lambda \cdot \Omega(\theta) \qquad \text{with} \qquad \Omega(\theta) = \text{KL}(\hat{p} \,\|\, p_\theta) + \eta \cdot (\|\boldsymbol{a}\|^2 + \sum_j \|\boldsymbol{w}_j\|^2)$$

starting at point $\theta_0 = \arg\min_{\theta \in \Theta} \Omega(\theta)$, and where $\lambda, \eta$ are two regularization hyperparameters that must be selected. Determining the starting point $\theta_0$ is analogous to having an initial pretraining phase. Thus, the proposed Wasserstein procedure can also be seen as finetuning a standard RBM, and forcing the finetuning not to deviate too much from the pretrained solution.

## 4 Experiments

We perform several experiments that demonstrate that Wasserstein-trained RBMs learn distributions that are better from a metric perspective. First, we explore what are the main characteristics of a learned distribution that optimizes the Wasserstein objective. Then, we investigate the usefulness of these learned models on practical problems, such as data completion and denoising, where the metric between observations occurs in the performance evaluation. We consider three datasets: *MNIST-small*, a subsampled version of the original MNIST dataset [11] with only the digits "0" retained, a subset of the UCI *PLANTS* dataset [19] containing the geographical spread of plants species, and *MNIST-code*, 128-dimensional code vectors associated to each MNIST digit (additional details in the supplement).

### 4.1 Training, Validation and Evaluation

All RBM models that we investigate are trained in full batch mode, using for $\hat{p}_\theta$ the PCD approximation [18] of $p_\theta$, where the sample is refreshed at each gradient update by one step of alternate Gibbs sampling, starting from the sample at the previous time step. We choose a PCD sample of same size as the training set ($N = \widetilde{N}$). The coefficients $\alpha_1, \ldots, \alpha_{\widetilde{N}}$ occurring in the Wasserstein gradient are obtained by solving the smoothed Wasserstein dual between $\hat{p}$ and $\hat{p}_\theta$, with smoothing parameter $\gamma = 0.1$ and distance $D(\boldsymbol{x}, \boldsymbol{x}') = \mathcal{H}(\boldsymbol{x}, \boldsymbol{x}')/\langle \mathcal{H}(\boldsymbol{x}, \boldsymbol{x}')\rangle_{\hat{p}}$, where $\mathcal{H}$ denotes the Hamming distance between two binary vectors. We use the centered parameterization of the RBM for gradient descent [13, 3]. We perform holdout validation on the quadratic containment coefficient $\eta \in \{10^{-4}, 10^{-3}, 10^{-2}\}$, and on the KL weighting coefficient $\lambda \in \{0, 10^{-1}, 10^0, 10^1, \infty\}$. The number of hidden units of the RBM is set heuristically to 400 for all datasets. The learning rate is set heuristically to $0.01(\lambda^{-1})$ during the pretraining phase and modified to $0.01\min(1, \lambda^{-1})$ when training on the final objective. The Wasserstein distance $\mathcal{W}_\gamma(\hat{p}_\theta, \hat{p})$ is computed between the whole test distribution and the PCD sample at the end of the training procedure. This sample is a fast approximation of the true unbiased sample, that would otherwise have to be generated by annealing or enumeration of the states (see the supplement for a comparison of PCD and AIS samples).

### 4.2 Results and Analysis

The contour plots of Figure 2 show the effect of hyperparameters $\lambda$ and $\eta$ on the Wasserstein distance. For $\lambda = \infty$, only the KL regularizer is active, which is equivalent to minimizing a standard RBM. As we reduce the amount of regularization, the Wasserstein distance becomes effectively minimized and thus smaller. If $\lambda$ is chosen too small, the Wasserstein distance increases again, for the stability reasons mentioned in Section 3. In all our experiments, we observed that KL pretraining was necessary in order to reach low Wasserstein distance. Not doing so leads to degenerate solutions. The relation between hyperparameters and minimization criteria is consistent across datasets: In all cases, the Wasserstein RBM produces lower Wasserstein distance than a standard RBM.

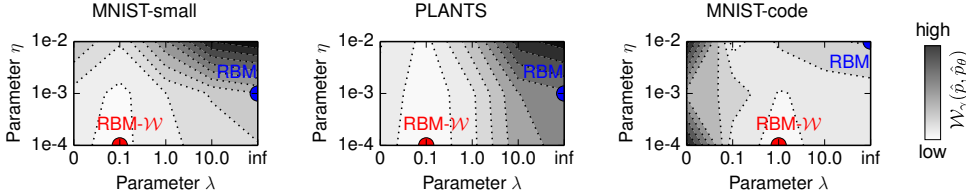

Figure 2: Wasserstein distance as a function of hyperparameters $\lambda$ and $\eta$. The best RBMs in the Wasserstein sense (RBM-$\mathcal{W}$) are shown in red. The best RBMs in the standard sense (i.e. with $\lambda$ forced to +inf, and minimum KL) are shown in blue.

Samples generated by the standard RBM and the Wasserstein RBM (more precisely their PCD approximation) are shown in Figure 3. The RBM-$\mathcal{W}$ produces a reduced set of clean prototypical examples, with less noise than those produced by a regular RBM. All zeros generated by RBM-$\mathcal{W}$

have well-defined contours and a round shape but do not reproduce the variety of shapes present in the data. Similarly, the plants territorial spreads generated by the RBM-$\mathcal{W}$ form compact and contiguous regions that are prototypical of real spreads, but are less diverse than the data or the sample generated by the standard RBM. Finally, the RBM-$\mathcal{W}$ generates codes that, when decoded, are closer to actual MNIST digits.

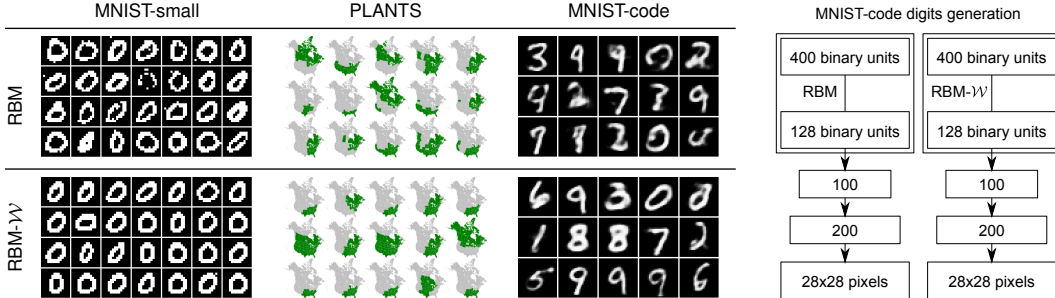

Figure 3: Examples generated by the standard and the Wasserstein RBMs. (Images for PLANTS dataset are automatically generated from the Wikimedia Commons template `https://commons.wikimedia.org/wiki/File:BlankMap-USA-states-Canada-provinces.svg` created by user Lokal_Profil.) Images for MNIST-code are produced by the decoders shown on the right.

The PCA plots of Figure 4 superimpose to the true data distribution (in gray) the distributions generated by the standard RBM (in blue) and the Wasserstein RBM (in red). In particular, the plots show the projected distributions on the first two PCA components of the true distribution. While the standard RBM distribution uniformly covers the data, the one generated by the RBM-$\mathcal{W}$ consists of a finite set of small dense clusters that are scattered across the input distribution. In other words, the Wasserstein model is biased towards these clusters, and systematically ignores other regions.

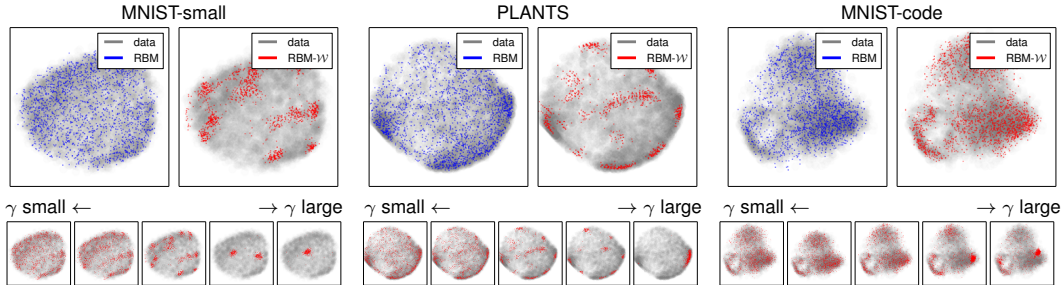

Figure 4: Top: Two-dimensional PCA comparison of distributions learned by the RBM and the RBM-$\mathcal{W}$ with smoothing parameter $\gamma = 0.1$. Plots are obtained by projecting the learned distributions on the first two components of the true distribution. Bottom: RBM-$\mathcal{W}$ distributions obtained by varying the parameter $\gamma$.

At the bottom of Figure 4, we analyze the effect of the Wasserstein smoothing parameter $\gamma$ on the learned distribution, with $\gamma = 0.025, 0.05, 0.1, 0.2, 0.4$. We observe on all datasets that the stronger the smoothing, the stronger the shrinkage effect. Although the KL-generated distributions shown in blue may look better (the red distribution strongly departs visually from the data distribution), the red distribution is actually superior if considering the smooth Wasserstein distance as a performance metric, as shown in Figure 2.

### 4.3 Validating the Shrinkage Effect

To verify that the shrinkage effect observed in Figure 4 is not a training artefact, but a truly expected property of the modeled distribution, we analyze this effect for a simple distribution for which the parameter space can be enumerated. Figure 5 plots the Wasserstein distance between samples of size 100 of a 10-dimensional Gaussian distribution $p \sim \mathcal{N}(0, I)$, and a parameterized model of that distribution $p_\theta \sim \mathcal{N}(0, \theta^2 I)$, where $\theta \in [0, 1]$. The parameter $\theta$ can be interpreted as a shrinkage

parameter. The Wasserstein distance is computed using the cityblock or euclidean metric, both rescaled such that the expected distance between pairs of points is 1.

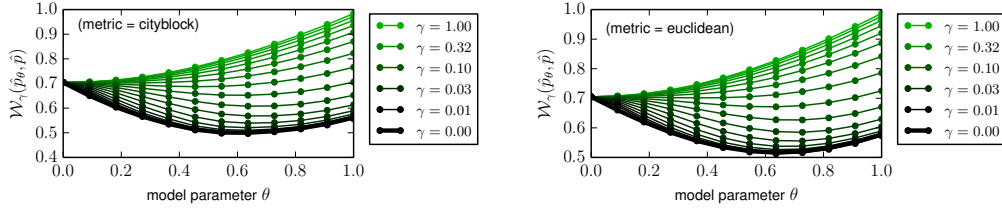

Figure 5: Wasserstein distance between a sample $\hat{p} \sim \mathcal{N}(0, I)$, and a sample $\hat{p}_\theta \sim \mathcal{N}(0, \theta^2 I)$ for various model parameters $\theta \in [0, 1]$ and smoothing $\gamma$, using the cityblock or the euclidean metric.

Interestingly, for all choices of Wasserstein smoothing parameters $\gamma$, and even for the true Wasserstein distance ($\gamma = 0$, computed here using the `OpenCV` library), the best model $p_\theta$ in the empirical Wasserstein sense is a shrinked version of $p$ (i.e. with $\theta < 1$). When the smoothing is strong enough, the best parameter becomes $\theta = 0$ (i.e. Dirac distribution located at the origin). Overall, this experiment gives a training-independent validation for our observation that Wasserstein RBMs learn shrinked cluster-like distributions. Note that the finite sample size prevents the Wasserstein distance to reach zero, and always favors shrinked models.

## 4.4 Data Completion and Denoising

In order to demonstrate the practical relevance of Wasserstein distance minimization, we apply the learned models to the task of data completion and data denoising, for which the use of a metric is crucial: Data completion and data denoising performance is generally measured in terms of *distance* between the true data and the completed or denoised data (e.g. Euclidean distance for real-valued data, or Hamming distance $\mathcal{H}$ for binary data). Remotely located probability mass that may result from simple KL minimization would incur a severe penalty on the completion and denoising performance metric. Both tasks have useful practical applications: Data completion can be used as a first step when applying discriminative learning (e.g. neural networks or SVM) to data with missing features. Data denoising can be used as a dimensionality reduction step before training a supervised model. Let the input $\boldsymbol{x} = [\boldsymbol{v}, \boldsymbol{h}]$ be composed of $d - k$ visible variables $\boldsymbol{v}$ and $k$ hidden variables $\boldsymbol{h}$.

**Data Completion** The setting of the data completion experiment is illustrated in Figure 6 (top). The distribution $p_\theta(\boldsymbol{x}|\boldsymbol{v})$ over possible reconstructions can be sampled from using an alternate Gibbs sampler, or by enumeration. The expected Hamming distance between the true state $\boldsymbol{x}^\star$ and the reconstructed state modeled by the distribution $p_\theta(\boldsymbol{x}|\boldsymbol{v})$ is given by iterating on the $2^k$ possible reconstructions:

$$\mathcal{E} = \sum_{\boldsymbol{h}} p_\theta(\boldsymbol{x} \,|\, \boldsymbol{v}) \cdot \mathcal{H}(\boldsymbol{x}, \boldsymbol{x}^\star)$$

where $\boldsymbol{h} \in \{0, 1\}^k$. Since the reconstruction is a probability distribution, we can compute the expected Hamming error, but also its bias-variance decomposition. On MNIST-small, we hide randomly located image patches of size $3 \times 3$ (i.e. $k = 9$). On PLANTS and MNIST-code, we hide random subsets of $k = 9$ variables. Results are shown in Figure 7 (left), where we compare three types of models: Kernel density estimation (KDE), standard RBM (RBM) and Wasserstein RBM (RBM-$\mathcal{W}$). The KDE estimation model uses a Gaussian kernel, with the Gaussian scale parameter chosen such that the KL divergence of the model from the validation data is minimized. The RBM-$\mathcal{W}$ is better or comparable the other models. Of particular interest is the structure of the expected Hamming error: For the standard RBM, a large part of the error comes from the variance (or entropy), while for the Wasserstein RBM, the bias term is the most contributing. This can be related to what is observed in Figure 4: For a data point outside the area covered by the red points, the reconstruction is systematically redirected towards the nearest red cluster, thus, incurring a systematic bias.

**Data Denoising** Here, we consider a simple noise process where for a predefined subset of $k$ variables, denoted by $\boldsymbol{h}$ a known number $l$ of bits flips occur randomly. Remaining $d - k$ variables are denoted by $\boldsymbol{v}$. The setting of the experiment is illustrated in Figure 6 (bottom). Calling $\boldsymbol{x}^\star$ the original and $\widetilde{\boldsymbol{x}}$ its noisy version resulting from flipping $l$ variables of $\boldsymbol{h}$, the expected Hamming error

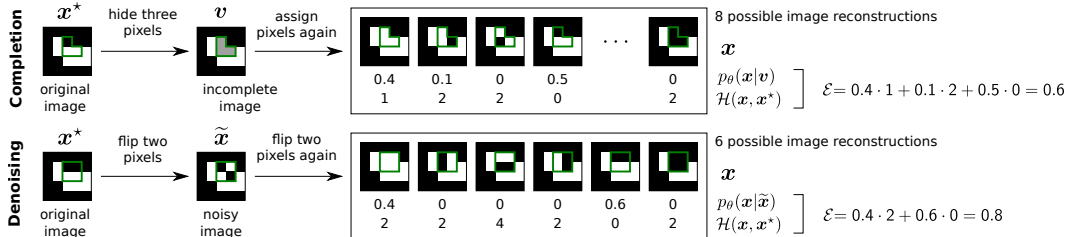

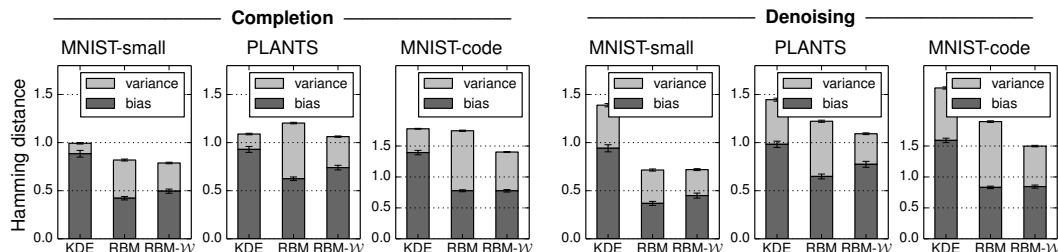

Figure 6: Illustration of the completion and denoising setup. For each image, we select a known subset of pixels, that we hide (or corrupt with noise). Each possible reconstruction has a particular Hamming distance to the original example. The expected Hamming error is computed by weighting the Hamming distances by the probability that the model assigns to the reconstructions.

Figure 7: Performance on the completion and denoising tasks of the kernel density estimation, the standard RBM and the Wasserstein RBM. The total length of the bars is the expected Hamming error. Dark gray and light gray sections of the bars give the bias-variance decomposition.

is given by iterating over the $\binom{k}{l}$ states $x$ with same visible variables $v$ and that are at distance $l$ of $\widetilde{x}$:

$$\mathcal{E} = \sum_{h} p_\theta(x \mid v, \mathcal{H}(x, \widetilde{x}) = l) \cdot \mathcal{H}(x, x^\star)$$

where $h \in \{0, 1\}^k$. Note that the original example $x^\star$ is necessarily part of this set of states under the noise model assumption. For the MNIST-small data, we choose randomly located images patches of size $4 \times 3$ or $3 \times 4$ (i.e. $k = 12$), and generate $l = 4$ random bit flips within the selected patch. For PLANTS and MNIST-code, we generate $l = 4$ bit flips in $k = 12$ randomly preselected input variables. Figure 7 (right) shows the denoising error in terms of expected Hamming distance on the same datasets. The RBM-$\mathcal{W}$ is better or comparable to other models. Like for the completion task, the main difference between the two RBMs is the bias/variance ratio, where again the Wasserstein RBM tends to have larger bias. This experiment has considered a very simple noise model consisting of a fixed number of $l$ random bit flips over a small predefined subset of variables. Denoising highly corrupted complex data will however require to combine Wasserstein models with more flexible noise models such as the ones proposed by [17].

## 5 Conclusion

We have introduced a new objective for restricted Boltzmann machines (RBM) based on the smooth Wasserstein distance. We derived the gradient of the Wasserstein distance from its dual formulation, and used it to effectively train an RBM. Unlike the usual Kullback-Leibler (KL) divergence, our Wasserstein objective takes into account the metric of the data. In all considered scenarios, the Wasserstein RBM produced distributions that strongly departed from standard RBMs, and outperformed them on practical tasks such as completion or denoising.

More generally, we demonstrated empirically, that when learning probability densities, the reliance on distributions that incorporate indirectly the desired metric can be substituted for training procedures that make the desired metric directly part of the learning objective. Thus, Wasserstein training can be seen as a more direct approach to density estimation than regularized KL training. Future work will aim to further explore the interface between Boltzmann learning and Wasserstein minimization, with the aim to scale the newly proposed learning technique to larger and more complex data distributions.

## Acknowledgements

This work was supported by the Brain Korea 21 Plus Program through the National Research Foundation of Korea funded by the Ministry of Education. This work was also supported by the grant DFG (MU 987/17-1). M. Cuturi gratefully acknowledges the support of JSPS young researcher A grant 26700002. Correspondence to GM, KRM and MC.

## Footnotes

[2]When using the MLE principle, metric considerations play a key role to define densities $p_\theta$, e.g. the reliance of Gaussian densities on Euclidean distances. This is the kind of metric we take for granted in this work.

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
