[Supplementary Material]

# Wasserstein Training of Restricted Boltzmann Machines

## (Supplementary Material)

**Grégoire Montavon**
Technische Universität Berlin
gregoire.montavon@tu-berlin.de

**Klaus-Robert Müller**
Technische Universität Berlin
klaus-robert.mueller@tu-berlin.de

**Marco Cuturi**
CREST - ENSAE, Université Paris-Saclay
marco.cuturi@ensae.fr

## A    Datasets used for the Experiments

In the experiments section of the paper, we have considered three datasets: The first one is the MNIST-small dataset, obtained from the original MNIST handwritten digits dataset [2] consisting of 60000 handwritten digits of size $28 \times 28$, by downsizing images to $14 \times 14$ pixels, binarizing them with the mean pixel value as threshold, and retaining only digits with class "0". There are 5923 such examples. The second dataset is based on the UCI PLANTS dataset [5], that associates to each plant species, a 70-dimensional binary vector indicating its presence or absence in each US state or Canadian province. For the purpose of producing a smooth-looking distribution with a tractable amount of examples, too frequently or infrequently occurring plants are discarded with probability $1 - e^{-(3 \cdot (\nu - 0.5))^6}$ where $\nu \in [0, 1]$ is the plant occurrence frequency. This results in a dataset of 6539 plants. The third dataset is the MNIST-code dataset, obtained by learning a deep autoencoder [1] from the original MNIST data. Our encoder is trained trained layer-wise, with the top layer forming a 128-dimensional binary code. We select 6000 codes randomly out of the 60000 produced, and give them as input distribution to the RBM. Each dataset is then randomly partitioned in three equal-sized subsets used for training, validation and test. Randomly selected examples drawn from the three datasets are shown in Figure 1.

Figure 1: Datasets learned by the RBMs. The last one (MNIST-code) is visualized after decoding.

## B    Effect of Training Hyperparameters on the KL Divergence

For each dataset, we have measured the effect of the regularization hyperparameters $\lambda$ and $\eta$ on the Wasserstein distance between the two distributions. The same experiment for the KL divergence is shown in Figure 2. The likelihood term of the KL divergence is evaluated by estimating the partition function $Z$ using annealed importance sampling (AIS) [3, 4] with 100 examples annealed in 1000 steps of increasingly small temperature differences. As expected, the KL divergence is lowest when the KL regularization is strong ($\lambda$ high). The KL divergence stays relatively low for most choices

of hyperparameters, except when $\lambda$ is close to zero, in which case the KL regularizer is effectively deactivated. In absence of KL regularization, the possibility that a data point is supported by zero modeled probability (which would cause the KL divergence to be infinity) is only prevented indirectly by the intrinsic properties of the Boltzmann machine probability functions.

Figure 2: KL divergence of the empirical distribution from the modeled distribution, for different choices of regularization parameters $\lambda$ and $\eta$.

## C   Comparison between PCD and AIS samples

Two possible empirical distributions approximating what the model has learned are given by the final state of the PCD chain, or the sample that results from the AIS procedure. While the second procedure is more closely related to the final distribution, it is computationally more expensive to produce and not necessarily reliable for all hyperparameters considered in this paper. In the paper, we showed the PCD final samples after training. In Figure 3 we show these samples next to the ones generated by the AIS procedure. Both samples are projected on the first two PCA components of the true distribution. Some differences can be found between both types of samples, especially for Wasserstein-based models with large $\gamma$, where the sampling procedure is no longer stable. Nevertheless, we can observe in both cases the same shrinkage effect of Wasserstein training and its dependence on the smoothing parameter $\gamma$.

Figure 3: PCA projection of PCD and AIS samples. Blue distributions are those that minimize the KL divergence. Red distributions are those that minimize the smoothed Wasserstein distance, with the same choices of smoothing parameter $\gamma$ as in the paper.