[Reviews · NeurIPS 2016]

Reviewer 1

Summary

The paper proposes training RBMs using the gamma-smoothed Wasserstein distance W_{gamma}(\hat{p}, p_{theta}) (eqs 3 and 4), where \hat{p} is the data distribution and p_{theta} is the model. They obtain derivatives of W_{gamma}(\hat{p}, p_{theta}) and train with gradient descent. However, the shrinkage of the Gaussian example in sec 4.3 (and the results in sec 4.2) gives serious concern wrt the consistency of Wasserstein training and this issue MUST be investigated more carefully before the paper is suitable for publication.

Qualitative Assessment

The paper refers to [2] and says that those authors proved statistical consistency. However, I am then surprised to see in section 4.3 that non-zero shrinkage is obtained (including for gamma=0) for the very simple case of modelling a N(0,I) distribution with N(0, sigma^2 I). What is going on here?? A failure of consistency would be a serious flaw in the formulation of a statistical learning criterion. Also in sec 3 (Stability and KL regularization) the authors say that at least for learning based on samples (\hat{p}_{theta}) that some regularization wrt the KL divergence is required. This clearly weakens the "purity" of the smoothed Wasserstein objective fn. Experiments (sec 4) are carried out on MNIST-small (the 0s from MNIST), a subset of the UCI PLANTS dataset, and 28-d binary codes (MNIST code). The results (as reported in lines 168-172 and Figs 3 and 4) seem to produce "compact and contiguous regions that are prototypical of real spread, but are less diverse than the data". This reinforces my belief above (relating to the Gaussian example insec 4.3) that minimizing the (smoothed) Wasserstein does not lead to a consistent density estimator. In sec 4.4 the authors try to "spin" these weakness into strenghts wrt data completion or denoising. Overall: The authors are to be congratulated on investigating an interesting alternative to standard KL-based training of RBMs. However, the shrinkage of the Gaussian example in sec 4.3 gives serious concern wrt the consistency of Wasserstein training and this issue MUST be investigated more carefully before the paper is suitable for publication. Other points: Title: the title over claims. The paper only discusses training *restricted* BMs. Intro l 7: RBMs do not need to be restricted to binary x's, see e.g. Exponential Family Harmoniums https://papers.nips.cc/paper/2672-exponential-family-harmoniums-with-an-application-to-information-retrieval.pdf Fig 5 would be much more readable using different colors for the different gammas. Also last entry should be gamma=0 (not OpenCV). You can explain in the text that you used OpenCV to compute this. === COMMENTS POST REBUTTAL ===== I have read the other reviewers comments and the rebuttal. I will remove my charge of a "fatal flaw" based on the point that the sample size in the experiments in sec 4.3 is small. However, the KL version of the experiment (where we fit a Gaussian with cov matrix theta^2 I to n = 100 samples drawn from a 10-d Gaussian) the values of theta I get from the max likelihood estimator are all pretty much in the range 0.95-1.05 (for about 10 repetitions). This should be compared with something around theta = 0.65 (estimated by eye from Fig 5(left)) for the Wasserstein (gamma=0) estimator. The convergence rate given by the authors of O(n^-1/(10+1)) for this simple problem is truly appalling, and presumably is in general O(n^-1/(D+1)) for D dimensions. This gives rise to serious concerns about the practicality of such a method. It would be helpful to see a study in this simple Gaussian case, showing e.g. how the theta that minimizes W_0 varies with n, although the rate given above suggests convergence will be painfully slow. Can you understand in such a case why there is a shrinkage bias? I note the response to R5 wrt lines 177-178 i.e. "the entropy regularizer contributes to the clustering effect [...]. It also appears for the true W distance in Fig 5." One could use say a KL matching criterion KL(\hat{p}||p_theta) along with an entropy penalty (no Wasserstein), and my understanding from the quote above is that this will cause clumping, whose virtues are extolled in sec 4.4 wrt data completion and denoising. (Note that due to the KL regularization in section 3 this is partly happening anyway in many of the experiments). I also agree with other reviwers who found the clarity of technical explanation poor, and suggested use of supp mat to help with this. Overall, I am not really in favour of accepting this paper (although I could live with acceptance). There is quite a lot of interesting work, but the issue where the clumping comes from (Wasserstein or entropic prior) needs to be better articulated and dissected. I believe some of claims wrt Wasserstein training are actually to do with the entropic prior, and this should be teased out by comparing against a KL + entropic prior construction.

Confidence in this Review

2-Confident (read it all; understood it all reasonably well)


Reviewer 2

Summary

This paper studies using Wasserstein Loss as the objective of learning generative models, with a focus on learning restricted Boltzmann Machines. It explored the effects of varying the strength of entropy regularization to the Wasserstein objective and its impact on image completion and denoising.

Qualitative Assessment

I like using Wasserstein loss as an alternative to minimizing KL and agree that it can be more robust in some settings. One limitation of this approach is the computational cost. Even with the recently developed optimization techniques, this approached was only tried on small toy datasets by the authors. It is interesting to see the effects of increasing the strength of entropy regularization (Figure 4). Can you give some intuition for why increasing lambda leads to models that are more concentrated in the image space? Why is E_{\hat{p}}[\alpha^*] = 0 but in Eqn 5, E_{p}[\alpha^*] not 0?

Confidence in this Review

2-Confident (read it all; understood it all reasonably well)


Reviewer 3

Summary

This paper proposes to train Boltzmann machines using a Wasserstein distance between data and model, rather than log likelihood. This is a neat idea, and is expressed clearly. Novel unsupervised training objectives are of potentially very high importance. I have concerns that, since the approach is sample based, it may scale particularly poorly with dimensionality and sample size. All experiments were on unusually small data, were heavily regularized with standard KL divergence, and only showed improvement in terms of Wasserstein-like measures of performance.

Qualitative Assessment

eq 4: sum over "x, x'", rather than "xx'" eq 4: I struggled a lot with this equation. Especially, it seems that it is only defined for gamma > 0, and it's very difficult to understand how the last term induces the correct dependencies. More explanatory text, and a change to stated gamma bound, would be good. 67: alpha* hasn't been defined 92-93: I suspect this approach, of measuring the distance between two distributions using samples from the two distributions, will suffer from exactly the same problems as Parzen window estimates (ie, sec 3.5 in L. Theis, A. van den Oord, and M. Bethge, A note on the evaluation of generative models, 2016 ). Specifically -- you need a number of samples which is exponential in the dimensionality of the space for the estimate to be reasonable, and when there are too few samples there will be a strong bias of the model towards the mean of the data distribution. 115: So this method would not be suitable for minibatch-training? 121-122: Why are local minima more of a problem for the Wasserstein case than the KL case? This didn't make sense to me. 137-139: All of these datasets are unusually small. Can you talk about the scaling properties of this algorithm, both in terms of number of samples and problem dimensionality? 177-178: It seems that the clustering is a result of the entropy regularizer, as opposed to the Wasserstein distance itself. sec 4.3: OK -- but is this shrinkage a desirable property? Will this be much worse for higher dimensional problems? I suspect it will. sec 4.4: All the performance measures are with respect to Wasserstein or Hamming distance. =========================== post rebuttal =========================== Thank you for your response! It clarified a lot. I would encourage you to include discussion of scaling in the final paper. (re "When γ=0, we recover the usual OT dual constraints.", I think this should rather be in the limit as γ goes to 0?)

Confidence in this Review

2-Confident (read it all; understood it all reasonably well)


Reviewer 4

Summary

This paper introduced a new objective function for restricted Boltzmann machine by replacing KL divergence with smoothed Wasserstien distance, resulting in a generative model suitable for problems where the use of metric is important. The paper provided theoretical proof for the derivation of the gradient for the new objective function, quantitative comparisons for analyzing the characteristics of the smoothed Wasserstein distance, and applications in data completion and desnoising to demonstrate the usefulness of this new objective function. Generally speaking, this paper is well written in respect of language expression and internal logic structure.

Qualitative Assessment

This is paper provided solid prove, well designed experiments and potential application scenarios. The description of the idea is easy to understand and the motivation is clearly demonstrated, i.e. the KL divergence is not good enough to act as a metric for problems where performance is measured in terms of distance. However, the comparison of the results only conducted among the kernal density estimation, standard RBM and Waserstein RBM, where the previous two both use non-distance metrics. It would be interesting to include the results of Euclidean distance and Hamming distance for comparison.

Confidence in this Review

1-Less confident (might not have understood significant parts)


Reviewer 5

Summary

This paper proposes to use Wasserstein distance instead of the commonly-used KL divergence as the cost function of RBMs. It presents advantages of Wasserstein distance through toy examples and promising experimental results.

Qualitative Assessment

I am convinced on the good direction this paper perused to apply Wasserstein distance to machine learning, and on the advantages of Wasserstein RBM which considers the metric structure of {0,1}^d while classical RBM does not. My main critism is regarding the writing. It is not self-closed with many outgoing links and requires one without prior knowledge of Wasserstein distance to read the refernce [4][5]. It seems that the author(s) didn't put much efforts on making it easier to read and broadening the potential audience. For example, it is useful to introduce, in the RBM setting, the optimal transport problem after equation 3, and then the dual problem. then the optimal dual alpha^\star. In its current form, it is not clear how alpha^\star is obtained. Similarly, in equation 6, it is not useful to point the reader to [5] without explaining the expression of alpha^\star(\tilde{x}_n) and its meaning. The author(s) are suggested to rewrite section 2 to be more self-closed and to be more intuitive. To save space the author(s) can put the proofs into supplementary and streamline section 3. As an unsupervised learning method, the proposed Wasserstein RBM has three additional hyper-parameters: gamma in equation 3, and lambda and eta in the end of page 4. The requirement of tuning these hyper-parameters puts a limit on the usefulness of the proposed method. As the author(s) said, "the proposed procedures can be seen as fine-tuning a standard RBM". Is lambda and eta necessary because they can give an improved energy surface that is easier to optimize, or because that without them the minima of the cost function is essentially different and gives trivial solutions? The Wasserstein distance is well defined on the parameter manifold including its boundaries, while KL divergence is not. This has a meaning in de-noising problems. If certain input dimension is mostly 0's with only a few 1's , a Wasserstein RBM can remove the noise and make the dimension deterministic, while KL-based RBM cannot. Do similar observations appear in the experiments? This might be a potential advantage of the proposed method. Minor comments: The title could be misunderstood as improving the training of the classical RBM. However the method proposes a different cost function and essentially is a different learning method. Equation 4. It is easier to read if the sum is over (x, x') instead of (xx'). Line 91: why the dual potential is centered? This is not straightforward and need more words. Section 3: the beginning has some overlapping contents with section 1 and can be streamlined.

Confidence in this Review

2-Confident (read it all; understood it all reasonably well)


Reviewer 6

Summary

The paper introduces the idea of augmenting the standard MLE objective for training an RBM with a smooth Wasserstein distance. The Wassterstein distance is a function of a chosen metric on the data space, and as such encourages the model to put mass in areas not far from the data according to the metric. This results in more plausible samples from the model at the possible expense of coverage.

Qualitative Assessment

The paper is clear and well-written. The clustering or shrinkage property observed in the experiments is compelling and could be very useful if the method can be scaled to problems of practical interest. On this note I think it would be useful to include details on how long experiments took to run, and more generally some information about complexity of the steps involved would be nice. Objective functions that induce more realistic samples is currently a hot topic in the variational autoencoder / generative adversarial network community, and I think that this work makes a useful contribution along related lines for the RBM. If there were possible applications to those methods as well it would be well-worth exploring.

Confidence in this Review

2-Confident (read it all; understood it all reasonably well)